# Associations between the *C3orf20* rs12496846 Polymorphism and Both Postoperative Analgesia after Orthognathic and Abdominal Surgeries and *C3orf20* Gene Expression in the Brain

**DOI:** 10.3390/pharmaceutics14040727

**Published:** 2022-03-28

**Authors:** Daisuke Nishizawa, Makoto Nagashima, Shinya Kasai, Junko Hasegawa, Kyoko Nakayama, Yuko Ebata, Ken-ichi Fukuda, Tatsuya Ichinohe, Masakazu Hayashida, Kazutaka Ikeda

**Affiliations:** 1Addictive Substance Project, Tokyo Metropolitan Institute of Medical Science, Tokyo 156-8506, Japan; nishizawa-ds@igakuken.or.jp (D.N.); kasai-sy@igakuken.or.jp (S.K.); hasegawa-jk@igakuken.or.jp (J.H.); nakayama-kk@igakuken.or.jp (K.N.); ebata-yk@igakuken.or.jp (Y.E.); mhaya@juntendo.ac.jp (M.H.); 2Department of Surgery, Toho University Sakura Medical Center, Sakura 143-8541, Japan; nagashima@sakura.med.toho-u.ac.jp; 3Division of Special Needs Dentistry and Orofacial Pain, Department of Oral Health and Clinical Science, Tokyo Dental College, Tokyo 101-0061, Japan; kfukuda@tdc.ac.jp; 4Department of Dental Anesthesiology, Tokyo Dental College, Tokyo 101-0061, Japan; ichinohe@tdc.ac.jp; 5Department of Anesthesiology, Saitama Medical University International Medical Center, Saitama 350-1298, Japan; 6Department of Anesthesiology and Pain Medicine, Juntendo University School of Medicine, Tokyo 113-8421, Japan

**Keywords:** opioids, analgesics, single-nucleotide polymorphisms, orthognathic surgery, abdominal surgery

## Abstract

Considerable individual differences are widely observed in the sensitivity to opioid analgesics. We focused on rs12496846, rs698705, and rs10052295 single-nucleotide polymorphisms (SNPs) in the *C3orf20*, *SLC8A2*, and *CTNND2* gene regions that we previously identified as possibly associated with postoperative analgesia after orthognathic surgery. We investigated associations between these SNPs and postoperative analgesia in 112 patients who underwent major open abdominal surgery in hospitals and were treated with analgesics, including opioids, after surgery. Total genomic DNA was extracted from peripheral blood or oral mucosa samples for genotyping each SNP. Effects of these potent SNPs on gene expression in the brain were also investigated in samples that were provided by the Stanley Foundation Brain Bank. In the association studies, carriers of the G allele of the rs12496846 SNP in the *C3orf20* gene region were significantly associated with greater 24 h postoperative analgesic requirements among the three SNPs that were investigated (*p* = 0.0015), which corroborated a previous study of orthognathic patients (*p* < 0.0001). In the gene expression analysis, carriers of the G allele of the rs12496846 SNP were significantly associated with lower mRNA expression of the *C3orf20* gene (*p* < 0.0001). These results indicate that this SNP could serve as a marker that predicts analgesic requirements.

## 1. Introduction

Opioids, such as morphine, codeine, oxycodone, and fentanyl, are widely used as effective analgesics for the treatment of acute and chronic pain because of their robust antinociceptive effects. However, their effects are not uniform across all patients. Considerable differences in the responsiveness or sensitivity to opioids are widely known [1,2]. This can impact analgesic effects that are required for adequate pain relief, which can hamper the effective clinical treatment of pain. The required amount of clinically prescribed opioid analgesics may also vary among patients with pain that is caused by malignant disease or surgery, depending on environmental factors, such as age, sex, weight, basal pain sensitivity, the type of surgery, perceived pain during the perioperative period [2], and genetic factors. According to twin studies of experimental heat and cold pressor pain by Angst et al. (2010, 2012) [3,4], genetic effects were estimated to account for 12%, 60%, and 30% of the observed response variance (i.e., pain threshold) after administration of the opioid analgesic alfentanil for heat pain, cold-pressor pain, and cold-pressor pain, respectively. However, these findings from twin studies did not provide information about potential genes that are involved in these responses.

In order to identify relationships between genetic variations, mostly single-nucleotide polymorphisms (SNPs), human opioid sensitivity, and related phenotypes, many candidate gene association studies have been conducted [5,6,7]. These studies have typically focused on genes that are involved in pharmacokinetic or pharmacodynamic opioidergic pathways or pain-related genes of various modalities. Target genes in these studies have included the μ-opioid receptor (*OPRM1*), cytochrome P450, family 2, subfamily D, polypeptide 6 (*CYP2D6*), adenosine triphosphate-binding cassette (ABC), subfamily B (MDR/TAP), member 1 (*ABCB1*), catechol-*O*-methyltransferase (*COMT*), and genes that are related to cytokines (e.g., interleukin-1β [IL-1β], IL-6, and tumor necrosis factor-α [TNF-α]), among others [2]. More details are shown elsewhere [5,6,7].

Genetic factors that are related to individual differences in the analgesic potency of opioids can also be explored using a genome-wide approach, namely genome-wide association studies (GWASs), that target genetic variations of all regions in the human genome regardless of preexisting assumptions about the phenotypes of interest [8]. However, only a few studies have conducted such investigations for human opioid sensitivity or responsiveness. One of these studies was a prospective cross-sectional multinational, multicenter study of patients with cancer from 11 European countries [9]. Patients were treated with opioids for moderate or severe pain, and the rs12948783 SNP, which maps to chromosome 17 upstream of the *RHBDF2* gene, showed the strongest association with responsiveness to opioids [10]. We conducted a GWAS of phenotypes that are related to opioid sensitivity, in which we recruited subjects who were scheduled to undergo cosmetic orthognathic surgery for mandibular prognathism [11]. A total of 9, 12, and 10 SNPs were selected as the top candidates that were associated with postoperative opioid analgesic requirements in additive, dominant, and recessive genetic models for each minor allele, respectively [12]. The best candidate in the additive model was the rs2952768 SNP, located near the *METTL21A* (*FAM119A*) and *CREB1* gene regions. This association was then replicated in patients who underwent major open abdominal surgery under combined general and epidural anesthesia [13]. However, replication studies of other candidate SNPs have been rarely conducted to date.

In the present study, we chose three SNPs among the list of the top candidate SNPs for opioid analgesia in the additive model (Appendix A). These three SNPs are located within the gene regions and not in flanking or intergenic regions. We then conducted a replication study to verify the findings of our previous GWAS.

## 2. Materials and Methods

### 2.1. Patients

#### 2.1.1. Patients Who Underwent Major Abdominal Surgery

The recruitment of subjects and basic protocol of postoperative pain management were described previously [13,14]. Briefly, the subjects in the association study included 112 patients who were Japanese and underwent major open abdominal surgery (28–80 years old, 60 males and 52 females). Subjects who underwent mostly gastrectomy for gastric cancer and colectomy for colorectal cancer under combined general and epidural anesthesia at the Research Hospital, Institute of Medical Science, The University of Tokyo, or Toho University Sakura Medical Center were included. Patients who were younger than 20 years old or older than 75 years old and patients with severe complications were excluded. The recruitment of subjects and analyses have been conducted since 2004. Peripheral blood or oral mucosa samples were collected from these subjects for gene analysis. Patients who underwent surgery without substantially severe pain (e.g., laparoscopy-assisted distal gastrectomy) were excluded from the analyses. Postoperative pain was managed primarily with continuous epidural analgesia with fentanyl or morphine. Whenever patients complained of significant postoperative pain despite continuous epidural analgesic, appropriate doses of opioids, including morphine, buprenorphine, pentazocine, and pethidine, or nonsteroidal anti-inflammatory drugs (NSAIDs), including diclofenac and flurbiprofen, were administered as rescue analgesics at the discretion of the surgeons. The clinical data that were collected included doses of rescue analgesics (opioids or NSAIDs) that were administered during the first 24 h postoperative period. To allow intersubject comparisons of rescue analgesic doses that were required during the first 24 h postoperative period, the doses of opioids and NSAIDs that were administered as rescue analgesics during this period were converted to the equivalent dose of systemic fentanyl according to previous reports [13,14]. The total dose of rescue analgesics that were administered was calculated as the sum of systemic fentanyl-equivalent doses of all opioids and NSAIDs that were administered to patients as rescue analgesics during the same period. The total dose of analgesics that were administered was calculated as the sum of the total dose of epidural analgesics and the total dose of rescue analgesics. Doses of analgesics that were administered postoperatively were normalized to body weight. The detailed demographic and clinical data of the subjects are detailed in Appendix A and previous reports [13,14].

The study was conducted according to guidelines of the Declaration of Helsinki and approved by the Institutional Review Board or Ethics Committee of Toho University Sakura Medical Center, Tokyo Dental College, and Tokyo Metropolitan Institute of Medical Science.

#### 2.1.2. Patients Who Underwent Painful Cosmetic Surgery

For the 355 Japanese patients who underwent painful cosmetic orthognathic surgery (i.e., mandibular sagittal split ramus osteotomy) for mandibular prognathism, the recruitment of subjects, surgical protocol, and subsequent postoperative pain management were fundamentally the same as in previous reports [11,12]. Patient-controlled fentanyl analgesia was continued for 24 h postoperatively with a CADD-Legacy patient-controlled analgesia (PCA) pump (Smiths Medical Japan, Tokyo, Japan). Postoperative PCA fentanyl use during this period was recorded. Doses of fentanyl that were administered postoperatively were normalized to body weight. The detailed demographic and clinical data of the subjects are provided in Appendix A and a previous report [12]. In previous studies, several SNPs have been identified to be associated with opioid analgesia in this cohort [11,12,15].

#### 2.1.3. Postmortem Specimens for Expression Analysis

In order to examine the mRNA expression levels of the *C3orf20* gene, postmortem human brain specimens were obtained from the Stanley Medical Research Institute (SMRI; Bethesda, MD, USA) [16] as samples that were independent of the samples in the association study of analgesic requirements as described previously [12]. The samples comprised a total of 105 human DNA samples that were extracted from the human occipital cortex and 100 RNA samples that were extracted from the human anterior cingulate cortex of the same specimens. The racial backgrounds of the subjects from which the samples for the study were obtained were 103 European Americans, one African American, and one Native American (19–64 years old, 69 males and 36 females). The other characteristics of the subjects were detailed in a previous report [16] and are provided on the SMRI website (https://www.stanleyresearch.org/brain-research/; accessed on 28 December 2021).

### 2.2. Genotyping Methods

In our previous GWAS that was conducted to identify potent SNPs that were associated with requirements for the opioid analgesic fentanyl during the 24 h postoperative period, nine SNPs were selected as the top candidates for the additive model for each minor allele. Among these, three SNPs (rs12496846, rs698705, and rs10052295) in the *C3orf20*, *SLC8A2*, and *CTNND2* gene regions, respectively, were selected for the present replication study because these SNPs were located within the gene regions, whereas the other six SNPs were located in flanking or intergenic regions.

Total genomic DNA was extracted from peripheral blood or oral mucosa samples using standard procedures or as described in a previous report [17]. The DNA concentration was adjusted to 5–50 ng/μL with TE buffer (10 mM Tris-HCl and 1 mM ethylenediaminetetraacetic acid [EDTA], pH 8.0) before use using a NanoDrop ND-1000 Spectrophotometer (NanoDrop Technologies, Wilmington, DE, USA). To genotype the *C3orf20* rs12496846, *SLC8A2* rs698705, and *CTNND2* rs10052295 SNPs, the TaqMan allelic discrimination assay (Life Technologies, Carlsbad, CA, USA) was used. A total of 112 DNA samples from patients who underwent major abdominal surgery were used for genotyping. Additionally, a total of 105 DNA samples from the postmortem specimens for the expression analysis were used for genotyping the rs12496846 SNP. For the SMRI samples, genomic DNA was extracted and adjusted to 10 ng/μL at SMRI.

To perform the TaqMan allelic discrimination assay with a LightCycler 480 (Roche Diagnostics, Basel, Switzerland), TaqMan SNP Genotyping Assays (Life Technologies, Carlsbad, CA, USA) that contained sequence-specific forward and reverse primers to amplify the polymorphic sequence and two probes that were labeled with VIC and FAM dye to detect both alleles of the rs12496846, rs698705, and rs10052295 SNPs (Assay ID: C___2077204_10, C___8714538_10, and C___2183581_10, respectively) were used. Real-time polymerase chain reaction (PCR) was performed in a final volume of 10 mL that contained 2× LightCycler 480 Probes Master (Roche Diagnostics, Basel, Switzerland), 40× TaqMan Gene Expression Assays, 5 ng genomic DNA as the template, and up to 10 mL H_2_O equipped with 2× LightCycler 480 Probes Master. The thermal conditions were the following: 95 °C for 10 min, followed by 45 cycles of 95 °C for 10 s and 60 °C for 60 s, with final cooling at 50 °C for 30 s. Afterward, endpoint fluorescence was measured for each sample well, and each genotype was determined based on the presence or absence of each type of fluorescence.

### 2.3. Real-Time Quantitative PCR

The SMRI RNA samples were treated as described previously [12]. First-strand cDNA for real-time quantitative PCR (qPCR) was synthesized with the SuperScriptIII First-Strand synthesis system for real-time qPCR (Life Technologies, Carlsbad, CA, USA) with 100 ng purified total RNA according to the manufacturer’s protocol.

To perform real-time qPCR with a LightCycler 480 (Roche Diagnostics, Basel, Switzerland), TaqMan Gene Expression Assays (Life Technologies, Carlsbad, CA, USA) were used as a probe/primer set that was specified for the *C3orf20* gene (Assay ID: Hs00297184_m1) and a probe/primer set for the *ACTB* gene (i.e., a housekeeping gene that encodes β-actin; Assay ID: Hs99999903_m1). The protocol and program for PCR were basically the same as in the previous report [12]. The expression level of the *C3orf20* gene was normalized to the expression level of the *ACTB* gene for each sample, and relative mRNA expression levels were compared between genotype subgroups. The experiments were performed in triplicate (separate experiments) for each sample, and average values were calculated for normalized expression levels.

### 2.4. Statistical Analysis

In the association studies of patients who underwent major abdominal surgery, the total dose of analgesics that were administered during the first 24 h postoperative period was used as an index of opioid sensitivity. Before the analyses, the quantitative values of total postoperative analgesic requirements (μg/kg) were natural-log-transformed for approximation to the normal distribution according to the following formula: *Value for analyses = Ln (1 + total postoperative analgesic requirements [μg/kg])*. In order to explore associations between the SNPs and phenotypes, linear regression analyses were conducted, in which total postoperative analgesic use (μg/kg; log-transformed) and genotype data for each SNP were incorporated as dependent and independent variables, respectively. Additive genetic models were used for the analyses to confirm the association that was observed in the previous GWAS study of painful cosmetic surgery [12]. The statistical analyses were performed using gPLINK v. 2.050, PLINK v. 1.07 (https://zzz.bwh.harvard.edu/plink/index.shtml; accessed on 28 December 2021) [18], and Haploview v. 4.2 [19]. The criterion for significance was set at *p* < 0.05, with Bonferroni correction for multiple comparisons as the post hoc test for the SNPs. Additionally, Hardy Weinberg equilibrium was tested using Exact Tests for genotypic distributions of the three SNPs [20]. The criterion for significance was set at a corrected *p* < 0.05. Overall, the association analysis in the present replication study was statistically the same as the analysis in our previous GWAS [12], although the surgery that the patients underwent and types of analgesics that were used were different between these two studies.

In the analysis of postmortem specimens for the expression analysis, the calculated expression level of the *C3orf20* gene normalized to the *ACTB* gene for each sample was used. Before the analysis, the quantitative values of the relative mRNA expression level were natural-log-transformed for approximation to the normal distribution according to the following formula: *Value for analysis = Ln (1 + relative expression level)*. In order to explore the association between the SNPs and phenotypes, linear regression analysis was performed, in which the relative expression level (log-transformed) and genotype data of the rs12496846 SNP were incorporated as dependent and independent variables, respectively. For the statistical analysis, SPSS 18.0J software (IBM, Armonk, NY, USA) was used. The criterion for significance was set at *p* < 0.05.

### 2.5. Additional In Silico Analysis

#### 2.5.1. Power Analysis

Statistical power analyses were preliminarily performed using G*Power 3.0.5 software [21]. Power analyses for the linear regression analysis revealed that the expected power (1 minus type II error probability) was 98.2% for a Cohen’s conventional “medium” effect size of 0.15 [22] when the type I error probability was set at 0.05 and the sample size was 112. However, for the same type I error probability and sample sizes of 112, the expected power decreased to 31.7% when Cohen’s conventional “small” effect size was 0.02. Conversely, the estimated effect size was 0.0713 for the same type I error probability and sample sizes of 112 to achieve 80% power. Therefore, a single analysis in the present study was expected to detect true associations with the phenotype with 80% statistical power for effect sizes from large to moderately small but not too small, although the exact effect size has been poorly understood in cases of SNPs that greatly contribute to opioid sensitivity.

#### 2.5.2. Linkage Disequilibrium Analysis

In order to identify relationships between the SNPs for the *C3orf20* gene region, linkage disequilibrium (LD) analysis was performed for a total of 127 samples from patients who underwent cosmetic orthognathic surgery [11,12] using Haploview v. 4.2 [19] for the locus of the ~136 kbp region that was annotated as the *C3orf20* gene and its flanking region on chromosome 3 based on an annotation file that was supplied by Illumina (San Diego, CA, USA). For the estimation of LD strength between SNPs, the commonly used *D′* and *r*^2^ values were pairwise-calculated using the genotype dataset of each SNP. Linkage disequilibrium blocks were defined as in the previous report [12].

#### 2.5.3. Reference of Databases

To further scrutinize the candidate SNP that may be associated with human opioid analgesic sensitivity, several databases, and bioinformatic tools were referenced, including the National Center for Biotechnology Information (NCBI) database (http://www.ncbi.nlm.nih.gov; accessed on 28 December 2021), Genotype-Tissue Expression (GTEx) portal (https://www.gtexportal.org/home/; accessed on 28 December 2021) [23], HaploReg v4.1 (https://pubs.broadinstitute.org/mammals/haploreg/haploreg.php; accessed on 28 December 2021) [24], and LDlink (https://ldlink.nci.nih.gov/?tab=home; accessed on 28 December 2021) [25]. The GTEx portal provides open access to such data as gene expression, quantitative trait loci, and histology images, based on the GTEx project, which is an ongoing effort to build a comprehensive public resource to study tissue-specific gene expression and regulation [23]. HaploReg is a tool for exploring annotations of the non-coding genome at variants on haplotype blocks, such as candidate regulatory SNPs at disease-associated loci [24]. LDlink is a suite of web-based applications that were designed to easily and efficiently interrogate LD in population groups [25].

## 3. Results

### 3.1. Replication of the Association between the C3orf20 rs12496846 SNP and Postoperative Analgesic Requirements in Patients Who Underwent Major Open Abdominal Surgery

To examine whether the candidate SNPs that were identified in our previous GWAS generally affect individual differences in opioid sensitivity, we sought to confirm the association between the *C3orf20* rs12496846, *SLC8A2* rs698705, and *CTNND2* rs10052295 SNPs and postoperative opioid requirements in another cohort of patients who underwent a different surgical procedure. The subjects who were recruited were 112 patients who underwent major open abdominal surgery under combined general and epidural anesthesia [13,14]. Genotype distributions of the three candidate SNPs are shown in Table 1. The distributions did not significantly deviate from the expected Hardy Weinberg equilibrium for all three SNPs (Table 1). A significant association was found between postoperative analgesic requirements and genotype of the *C3orf20* rs12496846 SNP (*β* = 0.1851, *p* = 0.0015; Table 2). In contrast, the *SLC8A2* rs698705 and *CTNND2* rs10052295 SNPs were not significantly associated with this phenotype (*β* = 0.0274, *p* = 0.7653, and *β* = −0.3412, *p* = 0.0546, respectively; Table 2). Analgesic requirements increased as the number of the G allele of the rs12496846 SNP that was carried in subjects increased (Figure 1), a pattern that was similar to subjects who underwent cosmetic orthognathic surgery (Appendix A). Total analgesic use, equipotent to systemic fentanyl, during the 24 h postoperative period was 2.090 ± 0.052, 2.281 ± 0.047, and 2.460 ± 0.060 μg/kg (log-transformed; mean ± SEM) in subjects with the A/A, A/G, and G/G genotypes, respectively.

### 3.2. Results of Linkage Disequilibrium Analysis for SNPs within/around the C3orf20 Gene

For SNPs in and around the *C3orf20* gene region, an LD analysis was conducted using genotype data from 126 samples in a total of 355 samples from subjects who underwent painful cosmetic surgery [11,12]. As a result, a total of 11 LD blocks (LD1-11) were observed within and around the gene region (Appendix A). The intronic rs12496846 SNP was in an LD block that includes several other intronic SNPs. In particular, this SNP was found to show strong LD with the rs12486391 SNP (*r^2^* = 0.92; Appendix A). No other SNPs were in strong LD (*r^2^* ≥ 0.80) with the rs12496846 SNP for the region investigated.

### 3.3. Association between the rs12496846 SNP and C3orf20 mRNA Expression Level

Considering the fact that the rs12496846 SNP is located in the intronic region of the *C3orf20* gene, one issue is the impact of this SNP on gene function or expression. As predicted by HaploReg v4.1 [24], this SNP could change three (i.e., HNF4, SP1, and Smad3) DNA motifs for DNA-binding proteins and could have regulatory effects on gene transcription. It suggests that expression of the *C3orf20* gene could be affected by this SNP, which might be related to a mechanism that contributes to individual differences in opioid sensitivity. Indeed, according to the GTEx portal, this SNP was shown to be significantly associated with *C3orf20* gene expression in the testis, *AC090952.5* gene expression in the testis, and *WNT7A* gene expression in the spinal cord (Appendix A) [23]. Although this SNP was not shown to be significantly associated with *C3orf20* gene expression in the brain according to the GTEx portal, one could expect that this SNP has some impact on gene expression in specific brain areas that are related to opioid analgesia. To pursue this issue, we examined the mRNA expression levels of the *C3orf20* gene using real-time qPCR with RNA samples that were extracted from the anterior cingulate cortex of postmortem subject specimens and compared mRNA expression levels between the genotype subgroups for the rs12496846 SNP, which were determined by genotyping the DNA samples that were extracted from the corresponding subjects. A significant association was found between the relative mRNA expression level of the *C3orf20* gene and genotype subgroups (*β* = −6.0418 × 10^−4^, *p* = 8.8301 × 10^−5^; Figure 2). The expression levels decreased as the number of the G allele of the rs12496846 SNP that was carried in subjects increased (Figure 2), a pattern that was similar to expression levels in the testis in the GTEx portal (Appendix A).

## 4. Discussion

In the present study, we focused on three SNPs in the *C3orf20*, *SLC8A2*, and *CTNND2* gene regions that we previously reported were possibly associated with postoperative analgesia after orthognathic surgery [12]. Although the roles of these genes have not been clarified in terms of opioid responses in previous studies except our own studies, we investigated associations between these SNPs and postoperative analgesia after abdominal surgery. The association was replicated only for the rs12496846 SNP of the *C3orf20* gene. The requirement of analgesics increased as the number of the G allele of the rs12496846 SNP that was carried in subjects increased (Figure 1). This trend was similar to subjects who underwent cosmetic orthognathic surgery (Appendix A). We also investigated the impact of this SNP on mRNA expression that was extracted from the anterior cingulate cortex of postmortem specimens. mRNA expression levels decreased as the number of the G allele of this SNP that was carried in subjects increased (Figure 2). Interestingly, the pattern was similar to expression levels in the testis in the GTEx portal (Appendix A). These results suggest that a part of the underlying mechanism by which the rs12496846 SNP affects opioid analgesia might be related to differences in mRNA expression of the *C3orf20* gene between genotypes of the rs12496846 SNP in the anterior cingulate cortex.

We identified an association between postoperative analgesia after abdominal surgery and the rs12496846 SNP, which was identified as a candidate in our previous GWAS of opioid analgesic requirements during the 24 h postoperative period in subjects who underwent painful cosmetic surgery. In the previous GWAS, nine, 12, and 10 SNPs were selected as the top candidates in the additive, dominant, and recessive genetic models, respectively [12]. The best candidate SNPs included several SNPs that mapped to 2q33.3–2q34 in the additive and recessive models, which were located near the *METTL21A* (*FAM119A*) and *CREB1* genes. One of the candidate SNPs was rs2952768, and consistent results were obtained for this SNP in patients who underwent abdominal surgery. Furthermore, another candidate SNP in the dominant model, rs1465040, close to the transient receptor potential subfamily C member 3 (*TRPC3*) gene, also exhibited an association with postoperative analgesia in patients who underwent abdominal surgery [15]. Therefore, associations with postoperative analgesia that were previously found in patients who underwent orthognathic surgery were replicated in the present study in patients who underwent abdominal surgery for the rs2952768, rs1465040, and rs12496846 SNPs based on similar statistical analyses. Although the rs2952768 and rs1465040 SNPs appear to not be related to the rs12496846 SNP in terms of principal functions of genes where the SNPs are located, the associations that were observed in these replication studies appear to suggest that the rs12496846, rs2952768, and rs1465040 SNPs all generally affect individual differences in opioid sensitivity, regardless of the type of surgery and perceived pain during the perioperative period in human subjects. We also showed that the rs12496846 SNP was associated with *C3orf20* mRNA expression levels, which is an additional novel finding in the present study.

The rs12496846 SNP was located in an LD block that includes several other intronic SNPs in the *C3orf20* gene region (Appendix A) on chromosome 3. The *C3orf20* gene encodes the chromosome 3 open reading frame 20 protein, although the characteristics and functions of its gene product are unknown. According to the NCBI database, the *C3orf20* gene is broadly expressed in the testis, bone marrow, and 17 other tissues, including the brain, and its gene product is located in the cytoplasm. In the GTEx portal, the rs12496846 SNP is mentioned as significantly associated with *C3orf20* gene expression in the testis, *AC090952.5* gene expression in the testis, and *WNT7A* gene expression in the spinal cord (Appendix A). The *AC090952.5* (*ENSG00000235629*) gene is also known as *LINC02922*, which encodes long intergenic non-protein coding RNA 2922, but the functions of this non-coding RNA are not characterized. The *WNT7A* gene encodes Wnt family member 7A protein and is a member of the WNT gene family, which consists of structurally related genes that encode secreted signaling proteins. Although these proteins have been implicated in oncogenesis and several developmental processes, including the regulation of cell fate and patterning during embryogenesis, the involvement of the *WNT7A* gene in the mechanism of opioid analgesia has not been clarified. Nevertheless, considering that the spinal cord plays an important role in the activation of descending pain modulatory circuits by opioids [26], the association between rs12496846 SNP genotypes and *WNT7A* gene expression levels in the spinal cord might have some implications in individual differences in opioid analgesia. The results of the present study showed that rs12496846 SNP genotypes might be implicated in individual differences in opioid analgesia, given that the anterior cingulate cortex is known to be one of the pain-modulating structures that are involved in opioid regulation, as well as the nucleus accumbens [27]. Future studies may further reveal detailed characteristics of the *C3orf20*, *LINC02922*, and *WNT7A* genes, the functions of which have not been previously clarified with regard to opioid analgesia.

One notable limitation of the present study was its relatively small sample size. However, a single analysis in the present study was expected to detect true associations with the phenotype with 80% statistical power for effect sizes from large to moderately small but not too small. Large sample sizes may not necessarily be required for pharmacogenomic studies relative to other kinds of studies, as previously stipulated [8].

There are scarce reports of genetic variations in the *C3orf20* gene that could affect some diseases or other phenotypic traits. Siuko et al. (2015) reported that four missense variants in the *C3orf20* gene were shared by two neuromyelitis optica patients [28]. According to the Phenotype-Genotype Integrator (PheGenI) in the NCBI database, other human genetic association studies have also suggested moderate associations between *C3orf20* SNPs and several phenotypes, including pancreatitis, precursor cell lymphoblastic leukemia-lymphoma, Sjogren’s syndrome, and Parkinson’s disease [29,30,31], although these associations have not been replicated and the underlying mechanisms remain unknown. Future studies will clarify the properties and functions of this gene and its genetic variations that could influence some human traits.

## 5. Conclusions

The results of the present study indicate that the rs12496846 SNP of the *C3orf20* gene could serve as a marker that predicts analgesic requirements, in which the G allele of this SNP is possibly associated with lower opioid sensitivity and thus greater requirements for opioid analgesics after painful cosmetic orthognathic surgery and major open abdominal surgery. Our findings provide valuable information for personalized pain treatment after both of these surgeries, in which the administration of more opioid analgesics may be needed for G-allele carriers of this SNP.

## Figures and Tables

**Figure 1 pharmaceutics-14-00727-f001:**
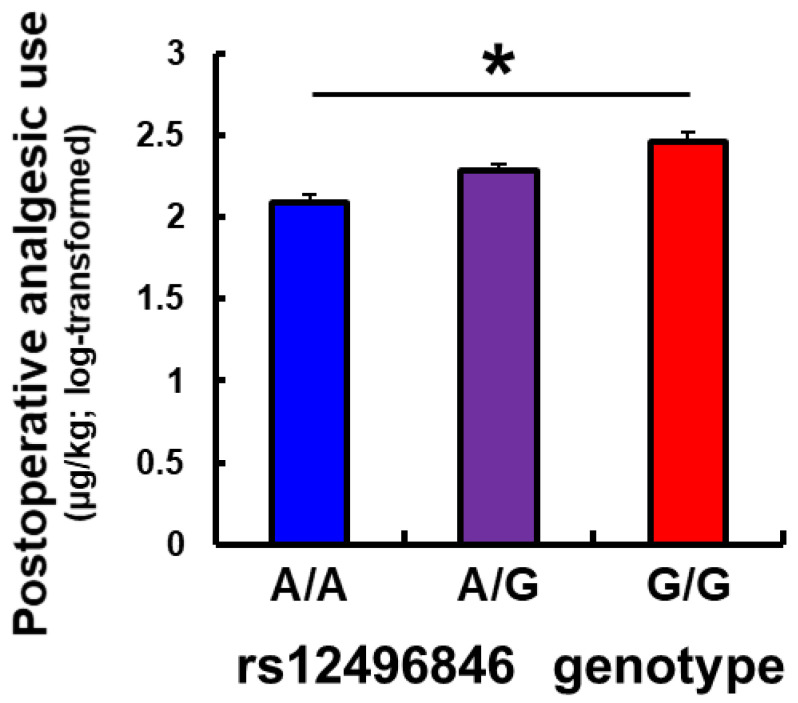
Association analysis between postoperative opioid analgesic requirements and the *C3orf20* rs12496846 SNP in subjects who underwent major open abdominal surgery, showing the total dose of analgesics that were administered per body weight (μg/kg; log-transformed) during the 24 h postoperative period. * Corrected *p* < 0.05, greater dose of analgesic administered as the number of the G allele of the rs12496846 SNP that was carried in subjects increased. The data are expressed as mean ± SEM.

**Figure 2 pharmaceutics-14-00727-f002:**
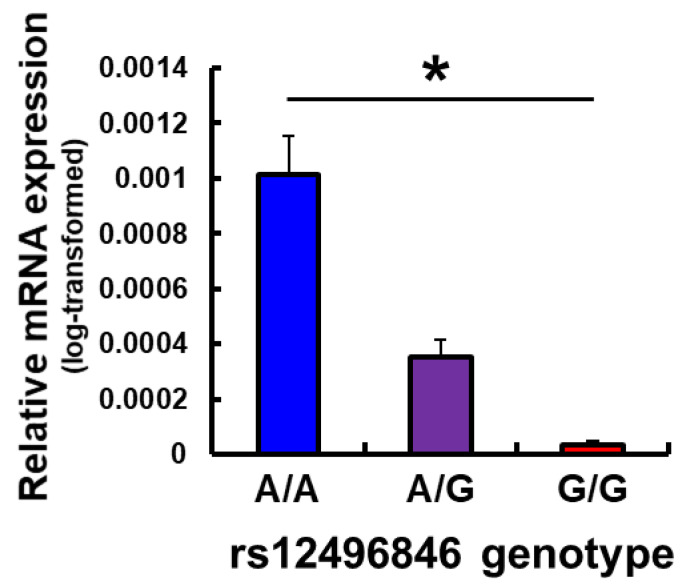
Relative mRNA expression level of the *C3orf20* gene between each genotype subgroup of the rs12496846 SNP in postmortem brains. * *p* < 0.05, lower level of mRNA expression as the number of the G allele of the rs12496846 SNP that was carried in subjects increased. The data are expressed as mean ± SEM.

**Table 1 pharmaceutics-14-00727-t001:** Genotype DISTRIBUTIONS of the three candidate SNPs in PATIENTS who underwent major open abdominal SURGERY.

Rank	SNP	Related Gene	Genotype (Frequency)			Allele (Frequency)		HWE Test
			AA	AB	AB	A	B	*p*
6	rs12496846	*C3orf20*	12 (0.1071)	55 (0.4911)	45 (0.4018)	79 (0.3527)	145 (0.6473)	0.5355
7	rs698705	*SLC8A2*	4 (0.0360)	19 (0.1712)	88 (0.7928)	27 (0.1216)	195 (0.8784)	0.05238
8	rs10052295	*CTNND2*	1 (0.0089)	22 (0.1964)	89 (0.7946)	24 (0.1071)	200 (0.8929)	1

CHR, chromosome number; Related gene, the nearest gene from the SNP site; A/A, homozygote for the minor allele in each SNP; A/B, heterozygote for the major allele in each SNP; B/B, homozygote for the major allele in each SNP; A, minor allele; B, major allele; HWE, Hardy-Weinberg equilibrium.

**Table 2 pharmaceutics-14-00727-t002:** Results of replication STUDY for three potent candidate SNPs that were selected in the GWAS (additive model).

Rank	SNP	CHR	Position	Related Gene	Location	GWAS		Replication Study	
						*β*	*p*	*β*	*p*
6	rs12496846	3	14748271	*C3orf20*	Intron	0.2431	3.241 × 10^−5^	0.1851	0.001527 *
7	rs698705	19	52629278	*SLC8A2*	Intron	−0.35	3.673 × 10^−5^	0.02739	0.7653
8	rs10052295	5	11030322	*CTNND2*	Intron	−0.4566	0.000199	−0.3412	0.05464

CHR, chromosome number; Position, chromosomal position (bp); Related gene, the nearest gene from the SNP site. * Corrected *p* < 0.05 after Bonferroni correction.

## Data Availability

The data that are presented in this study are available upon request from the corresponding author.

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
