# Peer review of "Associations between the C3orf20 rs12496846 Polymorphism and Both Postoperative Analgesia after Orthognathic and Abdominal Surgeries and C3orf20 Gene Expression in the Brain"

_pharmaceutics, 2022, doi:10.3390/pharmaceutics14040727_

Round 1

Reviewer 1 Report

This manuscript replicated the associations between C3orf20 rs12496846 polymorphism and opioid responses. Several questions are listed below:

1) Please describe the role of the studied genes, especially C3orf20, in opioid responses.

2) The sample size seems small, which leads to low statistical power. Did you perform the statistical power analysis? If not, please add the result of the post hoc power analysis.

3) Overall, the discussion section should be improved. The current discussion did not fully support your findings.

- (2nd paragraph) Two SNPs of rs2952768 and rs1465040 seem to have NO relation to rs12496846.

- (3rd and 4th paragraph) As you mentioned, the functions of related genes (e.g., C3orf20, LINC02922, WNT7A) have NOT been clarified.

4) Please add the comparisons between the original GWAS study and this replication study.

Minor comments:

1) Please add enough references (e.g., Ln39-40, Ln54-58, Ln64-68, Ln139, ...).

2) As you conducted association studies using two cohorts (patients who underwent major abdominal surgery and patients who underwent painful cosmetic surgery), please clarify the participants (including inclusion and exclusion criteria), setting (including study period), and study outcomes (-> especially for the second cohort). In addition, please provide the numbers of patients included in the study.

3) Are the replication cohorts independent from the previous GWAS?

4) Please report the ethnic composition of study cohorts.

5) For each SNP, did you test the Hardy Weinberg equilibrium?

6) Why do you perform the LD analysis using patients who underwent cosmetic orthognathic surgery (second cohort?), not using patients who underwent major abdominal surgery (first cohort?)?

7) If you investigated the same outcome, it is helpful to provide the pooled results using a meta-analysis approach.

8) Please provide the baseline characteristics of study populations.

Author Response

Response to Reviewer 1

Comments and Suggestions for Authors

This manuscript replicated the associations between C3orf20 rs12496846 polymorphism and opioid responses. Several questions are listed below:

1) Please describe the role of the studied genes, especially C3orf20, in opioid responses.

Response: None of the studied genes, including C3orf20, SLC8A2, and CTNND2, have been clarified in terms of their roles in opioid responses in previous reports. We added a sentence to describe this in the Discussion section [Lines 340-343].

2) The sample size seems small, which leads to low statistical power. Did you perform the statistical power analysis? If not, please add the result of the post hoc power analysis.

Response: According to the reviewer’s suggestion, we performed a statistical power analysis for the main study cohort (patients who underwent major open abdominal surgery) and reported the results of the power analysis in the Materials and Methods section (2.5. Additional in silico analysis, 2.5.1. Power analysis) [Lines 227-239].

3) Overall, the discussion section should be improved. The current discussion did not fully support your findings.

- (2nd paragraph) Two SNPs of rs2952768 and rs1465040 seem to have NO relation to rs12496846.

- (3rd and 4th paragraph) As you mentioned, the functions of related genes (e.g., C3orf20, LINC02922, WNT7A) have NOT been clarified.

Response: We expanded the Discussion according to the reviewer’s suggestions.

“Although the rs2952768 and rs1465040 SNPs appear to not be related to the rs12496846 SNP, these results suggest that the rs12496846, rs2952768, and rs1465040 SNPs all generally affect individual differences in opioid sensitivity, regardless of the type of surgery and perceived pain during the perioperative period in human subjects.”

“Future studies may further reveal detailed characteristics of the C3orf20, LINC02922, and WNT7A genes, the functions of which have not been previously clarified with regard to opioid analgesia” [Lines 364-368, 392-394].

4) Please add the comparisons between the original GWAS study and this replication study.

Response: According to the reviewer’s suggestion, we added a description of comparisons between the original GWAS study and this replication study in the Materials and Methods section.

“Overall, the association analysis in the present replication study was statistically the same as the analysis in our previous GWAS [12], although the surgery that the patients underwent and types of analgesics that were used were different between these two studies” [Lines 212-215].

Minor comments:

1) Please add enough references (e.g., Ln39-40, Ln54-58, Ln64-68, Ln139, ...).

Response: According to the reviewer’s suggestion, we cited additional references in the Introduction and Materials and Methods sections [Lines 42, 56, 67, 139, ...].

2) As you conducted association studies using two cohorts (patients who underwent major abdominal surgery and patients who underwent painful cosmetic surgery), please clarify the participants (including inclusion and exclusion criteria), setting (including study period), and study outcomes (-> especially for the second cohort). In addition, please provide the numbers of patients included in the study.

Response: According to the reviewer’s suggestion, we clarified the participants (including inclusion and exclusion criteria), setting (including the study period), and study outcomes (especially for the second cohort) in the Materials and Methods section [Lines 91-98]. We also provided the numbers of patients who were included in the study in the Materials and Methods section [Line 125].

3) Are the replication cohorts independent from the previous GWAS?

Response: The previous GWAS was conducted in patients who underwent cosmetic orthognathic surgery. The replication cohorts were patients who underwent major abdominal surgery. These were independent patient groups.

4) Please report the ethnic composition of study cohorts.

Response: The ethnic backgrounds of the study cohorts were all Japanese. According to the reviewer’s suggestion, we added the ethnic composition of the study cohorts in the Materials and Methods section [Lines 92, 125].

5) For each SNP, did you test the Hardy Weinberg equilibrium?

Response: We tested Hardy Weinberg equilibrium for each SNP in the main study cohort (patients who underwent major open abdominal surgery). We present the results in the new Table 1 [Lines 210-211, 273-275, 285-289].

6) Why do you perform the LD analysis using patients who underwent cosmetic orthognathic surgery (second cohort?), not using patients who underwent major abdominal surgery (first cohort?)?

Response: We performed linkage disequilibrium analysis using patients who underwent cosmetic orthognathic surgery (second cohort) because the subjects were recruited in a single hospital for this cohort, and genetic homogeneity would be relatively high compared with patients who underwent major abdominal surgery (first cohort) and were recruited in multiple hospitals in different locations.

7) If you investigated the same outcome, it is helpful to provide the pooled results using a meta-analysis approach.

Response: Although we investigated the same outcome, we did not consider that it would be suitable to provide the pooled results using a meta-analysis approach in the present study because some conditions of the patients who were included in the original GWAS and replication study were different (e.g., only fentanyl was used as the drug for analysis in our previous GWAS, whereas several drugs were used in the present replication study).

8) Please provide the baseline characteristics of study populations.

Response: According to the reviewer’s suggestion, we provided baseline characteristics of the two study populations (patients who underwent cosmetic orthognathic surgery and patients who underwent major abdominal surgery) in the new Supplementary Table S1 and Supplementary Table S2. These characteristics were also described in previous reports [references 12-14].

Reviewer 2 Report

The manuscript entitled „Associations between the C3orf20 rs12496846 polymorphism and both postoperative analgesia after orthognathic and abdominal surgeries and C3orf20 gene expression in the brain” presents three single-nucleotide polymorphisms in  correlation to postoperative analgesia in 112 patients underwent major open  abdominal surgery treated with analgesics.

My comments/suggestions:

  1. Line 83: In my opinion the simple graph with scheme of selection (based on lines 70-82) would systematize the methodolody of SNPs selection in this article.
  2. 1.2. Patients who underwent painful cosmetic surgery – add number of patients. 127 patients, right? As I  found it in 2.5.1. Linkage disequilibrium analysis there is subsection
  1. Lack of ethical approval information in Materials and methods section.
  2. Could you add the information about genotypes/alleles frequencies, please?
  3. Rearrange the abstract to make it more informative, please.
  4. Could you add limitations of this study, please?

Author Response

Response to Reviewer 2

Comments and Suggestions for Authors

The manuscript entitled „Associations between the C3orf20 rs12496846 polymorphism and both postoperative analgesia after orthognathic and abdominal surgeries and C3orf20 gene expression in the brain” presents three single-nucleotide polymorphisms in  correlation to postoperative analgesia in 112 patients underwent major open  abdominal surgery treated with analgesics.

My comments/suggestions:

  1. Line 83: In my opinion the simple graph with scheme of selection (based on lines 70-82) would systematize the methodolody of SNPs selection in this article.

Response: According to the reviewer’s suggestion, we added a graph that presents a schematic diagram of selection (based on lines 70-82) in Supplementary Figure S1 in the Introduction to systematize the methodolody of SNP selection.

  1. 1.2. Patients who underwent painful cosmetic surgery – add number of patients. 127 patients, right? As I found it in 2.5.1. Linkage disequilibrium analysis there is subsection

Response: According to the reviewer’s suggestion, we added the number of patients. The number of all patients in our previous GWAS was 355 [Line 126], and a portion of the patient sample was used for the linkage disequilibrium analysis.

  1. Lack of ethical approval information in Materials and methods section.

Response: According to the reviewer’s suggestion, we mentioned the ethics approval in the Materials and Methods section [Lines 121-124] and also at the end of the manuscript.

“The study was conducted according to guidelines of the Declaration of Helsinki and approved by the Institutional Review Board or Ethics Committee of Toho University Sakura Medical Center, Tokyo Dental College, and Tokyo Metropolitan Institute of Medical Science.”

  1. Could you add the information about genotypes/alleles frequencies, please?

Response: According to the reviewer’s suggestion, we added information about genotype/allele frequencies for each SNP in the main study cohort (patients who underwent major open abdominal surgery) and presented the data in the new Table 1.

  1. Rearrange the abstract to make it more informative, please.

Response: According to the reviewer’s suggestion, we included more information in the abstract.

“We focused on rs12496846, rs698705, and rs10052295 single-nucleotide polymorphisms (SNPs) in the C3orf20, SLC8A2, and CTNND2 gene regions that we previously identified as possibly associated with postoperative analgesia after orthognathic surgery. We investigated associations between these SNPs and postoperative analgesia in 112 patients who underwent major open abdominal surgery in hospitals and were treated with analgesics, including opioids, after surgery. Total genomic DNA was extracted from peripheral blood or oral mucosa samples for genotyping each SNP. Effects of these potent SNPs on gene expression in the brain were also investigated in samples that were provided by the Stanley Foundation Brain Bank” [Lines 21-28].

  1. Could you add limitations of this study, please?

Response: According to the reviewer’s suggestion, we added a paragraph that mentions some limitations of this study in the Discussion section.

“One notable limitation of the present study was its relatively small sample size. However, a single analysis in the present study was expected to detect true associations with the phenotype with 80% statistical power for effect sizes from large to moderately small but not too small. Large sample sizes may not necessarily be required for pharmacogenomic studies relative to other kinds of studies. Among GWASs, for example, several recent pharmacogenomic studies that involved around 100 cases have detected genome-wide significant associations, suggesting that at least some pharmacogenomic effects tend to be larger and involve fewer genes than those that are detected in GWASs for complex diseases [8]” [Lines 397-405].

Round 2

Reviewer 1 Report

Although the authors properly responded to the comment I suggested, I still have some more comments.

1) This study is not the GWAS study, but the replication study; please delete the sentences [Lines 399-403].

2) You mentioned that the rs2952768 and rs1465040 SNPs appear to not be related to the rs12496846 [Lines 364-365]. These two SNPs are about your previous study [12]; if two SNPs are not related to the present study, please delete the paragraph.

3) I'm wondering about the originality of this present study. Please clarify the differences between the previous study [12] and the present study.

Author Response

Response to Reviewers

Reviewer 1

Comments and Suggestions for Authors

Although the authors properly responded to the comment I suggested, I still have some more comments.

1) This study is not the GWAS study, but the replication study; please delete the sentences [Lines 399-403].

Response: According to the reviewer’s suggestion, we deleted this text [Line 408].

2) You mentioned that the rs2952768 and rs1465040 SNPs appear to not be related to the rs12496846 [Lines 364-365]. These two SNPs are about your previous study [12]; if two SNPs are not related to the present study, please delete the paragraph.

Response: Although the rs2952768 and rs1465040 SNPs appear to not be related to the rs12496846 SNP in terms of principal functions of genes where the SNPs are located, the associations that were observed in these replication studies appear to suggest that the rs12496846, rs2952768, and rs1465040 SNPs all generally affect individual differences in opioid sensitivity, regardless of the type of surgery and perceived pain during the perioperative period in human subjects. We revised the paragraph for clarity [Lines 355-377].

3) I'm wondering about the originality of this present study. Please clarify the differences between the previous study [12] and the present study.

Response: The previous study [12] was a GWAS that explored associations between hundreds of thousands of SNPs and postoperative analgesia in patients who underwent orthognathic surgery. The present study was conducted as a replication study that focused specifically on the rs12496846 SNP that was previously identified in our GWAS as a candidate SNP that may be associated with opioid analgesia [12]. Although common SNPs were analyzed in both studies using similar statistical methods, the subjects in the present study were separate patients who underwent abdominal surgery rather than orthognathic surgery. Furthermore, the present study also showed that the rs12496846 SNP was associated with C3orf20 mRNA expression levels, which is a novel additional finding in the present study. Therefore, we revised the text to further clarify differences between the previous study [12] and the present study [Lines 355-377].
